Temporal and spatial strategies in an active place avoidance task on Carousel: a study of effects of stability of arena rotation speed in rats

Bahník Štěpán 1 2 bahniks@seznam.cz
Stuchlík Aleš 1
1 Department of Neurophysiology of Memory, Institute of Physiology, Academy of Sciences of the Czech Republic , Prague , Czech Republic
2 Department of Psychology, University of Würzburg , Würzburg , Germany
Jimenez-Diaz Lydia
Electronic publication date: 2015 Sep 22
Publication date: 2015
Volume: 3
Electronic Location ID: e1257
Received 2015 May 6; Accepted 2015 Sep 1
Copyright: © 2015 Bahník and Stuchlík
Copyright year: 2015
Copyright holder: Bahník and Stuchlík
License: This is an open access article distributed under the terms of the Creative Commons Attribution License, which permits unrestricted use, distribution, reproduction and adaptation in any medium and for any purpose provided that it is properly attributed. For attribution, the original author(s), title, publication source (PeerJ) and either DOI or URL of the article must be cited.
License URL: https://creativecommons.org/licenses/by/4.0/

Keywords: Spatial navigation, Interval timing, Substratal idiothetic navigation, Inertial idiothetic navigation, Rats

Funding: MŠMT LH14053 RVO 67985823 Deutsche Forschungsgemeinschaft DFG-RTG 1253/2 The present study was funded mainly by grant MŠMT LH14053. Institutional support was covered by RVO: 67985823. The work of ŠB was partly funded by Deutsche Forschungsgemeinschaft (DFG-RTG 1253/2). The funders had no role in study design, data collection and analysis, decision to publish, or preparation of the manuscript.

==============================
The active place avoidance task is a dry-arena task used to assess spatial navigation and memory in rodents. In this task, a subject is put on a rotating circular arena and avoids an invisible sector that is stable in relation to the room. Rotation of the arena means that the subject’s avoidance must be active, otherwise the subject will be moved in the to-be-avoided sector by the rotation of the arena and a slight electric shock will be administered. The present experiment explored the effect of variable arena rotation speed on the ability to avoid the to-be-avoided sector. Subjects in a group with variable arena rotation speed learned to avoid the sector with the same speed and attained the same avoidance ability as rats in a group with a stable arena rotation speed. Only a slight difference in preferred position within the room was found between the two groups. No difference was found between the two groups in the dark phase, where subjects could not use orientation cues in the room. Only one rat was able to learn the avoidance of the to-be-avoided sector in this phase. The results of the experiment suggest that idiothetic orientation and interval timing are not crucial for learning avoidance of the to-be-avoided sector. However, idiothetic orientation might be sufficient for avoiding the sector in the dark.

Introduction

The active place avoidance task (APA; formerly also called active allothetic place avoidance) (Bures et al., 1997; Bures et al., 1998; Cimadevilla et al., 2000; Fenton et al., 1998; Stuchlik et al., 2007; Stuchlik, Petrasek & Vales, 2008; Stuchlik et al., 2013) is a variant of a place avoidance task (Bures et al., 1997) used for assessing spatial navigation and memory in rodents. The task uses a dry and smooth circular arena made of metal (Carousel) which contains an unmarked to-be-avoided sector (usually a 60° section of the arena), entering which is punished by a mild footshock. It can be used both for rats (Stuchlik, Petrasek & Vales, 2008; Lobellova et al., 2013; Zemanova et al., 2013) and mice (Burghardt et al., 2012; Vukovic et al., 2013). The to-be-avoided sector is stable within the room and the arena rotates on its axis in the active version of the task. This means that successful performance requires active avoidance of the to-be-avoided sector, e.g., to walk in the direction opposite to arena rotation. Since the sector is not directly visible, the subject has to remember its location in relation to orientation cues outside of the rotating arena, i.e., in the experiment room. Therefore, efficient avoidance of the sector requires continual locomotion (Stuchlik et al., 2013) and spatial navigation (Cimadevilla et al., 2000). The task and special modifications to it (such as two-frame place avoidance) were used in various domains of animal cognition studies such as in pharmacological (Prokopova et al., 2012; Rambousek et al., 2011), lesion (Cimadevilla, Fenton & Bures, 2001), genetic (Petrasek et al., 2014a; Petrasek et al., 2014b), and electrophysiological studies (Kelemen & Fenton, 2010). Notably, the task has been extensively employed in studies focused on animal models of neuropsychiatric disorders, involving schizophrenia (Lobellova et al., 2013; Stuchlik et al., 2004; Lee, Dvorak & Fenton, 2014), ischemia (Popp et al., 2011), and traumatic brain injury (Abdel Baki et al., 2009; Haber et al., 2013). Thus, the task has a high pre-clinical significance, despite the fact that only a few laboratories worldwide use it at this time. Therefore, attempts to provide deeper insight into learning mechanisms involved in performance in the task are of great importance (Blahna et al., 2011; Kubik, Stuchlik & Fenton, 2006).

This paper focuses on the examination of possible strategies to solve the task. The standard variant of the active place avoidance task on the Carousel uses a stable speed of arena rotation—usually 1 revolution per minute (rpm). One of the possible strategies of avoidance then is to move with the same average speed against the direction of arena rotation. In order to use this method of avoidance, a subject does not need to use allothetic orientation (using external spatial cues) (Wallace, Marting & Winter, 2008). Use of idiothetic orientation (based on information generated by locomotion of an animal) (Mauk & Buonomano, 2004), possibly with interval timing, i.e., perception of time at intervals ranging from seconds to minutes (Mittelstaedt & Mittelstaedt, 1980; Buhusi & Meck, 2005), would be sufficient in this case (Klement et al., 2010; Fajnerova et al., 2014). A subject may thus move continuously against the direction of arena rotation with the same average speed as the speed of arena rotation, or move against the direction of the rotation with a certain periodicity across the distance which regulates position of the subject within a room. If subjects, at least partly, use this avoidance strategy, the interpretation of results of the studies using the APA task would have to take into account that possible performance deficits may be caused by impairment in idiothetic orientation or interval timing. Interval timing is known to be influenced by drugs (Coull, Cheng & Meck, 2011) and its deficits are seen in neuropathology (Balci et al., 2009), which are both areas where the APA task is used (Stuchlik, Petrasek & Vales, 2008).

The present study used the manipulation of arena rotation speed to influence relevance of temporal information for successful avoidance in the APA task. Variable rotation speed should influence the possibility of using a combination of inertial idiothesis (i.e., use of information from internal organs sensing movement, such as proprioceptors, vestibular apparatus, and proprioception) (Mittelstaedt & Glasauer, 1991) and interval timing for avoidance of the to-be-avoided sector. On the other hand, in the case of stable arena rotation speed a subject can move with a certain frequency across distance needed for regulation of its position. For example, a subject on an arena with the stable rotation speed 1 rpm can move every 30 s by 180° to avoid the to-be-avoided sector. In the case of variable rotation speed, the subject has no such opportunity because the required distance and frequency of movement for regulation of its position within a room necessarily varies depending on rotation speed. Additionally, the use of inertial idiothesis may be further affected if rotation speed changes not only within a session but between sessions as well. In that case, a subject may not easily “learn the speed” required to regulate its position within a room. On the other hand, it is possible that variable arena rotation speed may influence attention given to inertial stimuli by making the rotation speed relevant for avoidance of the to-be-avoided sector and therefore making it salient for the subject, thus leading to better performance in the task. The present study allowed us to explore these possibilities.

Method

Ethics statement

This study was carried out in strict accordance with Animal Protection Code of Czech republic, EU directive 2010/63/EC and with the recommendations in the Guide for the Care and Use of Laboratory Animals of the National Institutes of Health. The protocol including needle implantation described in the Subjects section was approved by the Committee on the Ethics of Animal Experiments of the Institute of Physiology Academy of Sciences of the Czech Republic (Permit Number: 136/2013). No surgery was performed, and all efforts were made to minimize suffering of animals.

Subjects

The experiment was conducted with 16 male Long-Evans rats obtained from the breeding colony of the Institute of Physiology, Academy of Sciences of the Czech Republic. At the beginning of the experiment, the rats were 16–17 weeks old and had mean weight 415 g (SD = 27 g). They were housed in transparent plastic 25 × 30 × 40 cm boxes in an air-conditioned animal room with a stable temperature and 12/12h light/dark cycle. All parts of the experiment were conducted during the light phase of the day. Water and food pellets were available ad libitum throughout the study. Prior to behavioral tests, rats were implanted with hypodermic needle through skin fold between shoulders. The needle was used for attaching an alligator clip delivering electric shocks throughout the experiment. The implantation procedure is analogous to subcutaneous injection in humans and does not require anesthesia.

Design and procedure

The experiment consisted of a handling phase and four phases using the Carousel maze. At the beginning, subjects were handled three days for 5 min each day. Next, a 5-day habituation phase followed, during which rats were habituated to the experimental apparatus for 10 min each day. The arena was not rotating during the habituation phase. After habituation, subjects were divided into two groups (experimental and control) of 8 subjects such that their locomotion during the habituation phase was similar, but the assignment to groups was random.

The learning phase of the active place avoidance task was 9 days long and one 20 min session took place during each day for every rat. The session always began during the same time of a day for each rat. The two groups differed only in arena rotation speed. The speed was always 1 rpm and stable during the whole session for the control group and varied from 0.60 to 1.34 rpm for the experimental group. Depending on the day, the speed for the experimental group changed every one or two minutes or alternatingly after two and three minutes. The two speeds used for the experimental group during each session were chosen such that the average speed throughout the session was always 1 rpm (i.e., comparable to controls). The change of speed after one minute was used, at most, on two subsequent days to prevent the possible using of the change as temporal information and the speeds were not same any two subsequent days (more detail about rotation speeds can be found on osf.io/683xk/).

The following phase (hereafter probe phase) contained one 20 min long session of the active place avoidance task. A stable rotation speed of 1 rpm was used for both groups during this session. This phase was included to compare avoidance strategies using measures that may be dependent on rotation speed.

The last phase of the experiment (dark phase) was three days long and one session of the APA task was scheduled for each day. The sessions were conducted with a stable rotation speed 1 rpm in complete darkness. Subjects were therefore not able to use room orientation cues. Since rats may use not only visual room orientation cues, all possible olfactory cues were removed from the room and three air fresheners were attached to a Plexiglas arena wall to cover possible remaining cues. The Plexiglas itself further hindered use of olfactory and auditory room orientation cues. Additionally, the motor of the apparatus which is positioned under the centre of the arena is a source of loud noise and also prevents possible use of auditory cues.1 A summary of the phases of the experiment can be found in Table 1.

Table 1 Phases of the experiment.

Phase	Number of sessions	Speed of arena rotation	
		Control group	Experimental group	
Habituation	5	–	–	
Learning	9	1 rpm	0.60–1.34 rpm	
Probe	1	1 rpm	1 rpm	
Dark	3 (4)	1 rpm	1 rpm	

Apparatus

The Carousel maze (Fig. 1) consisted of a smooth metal circular arena 82 cm in diameter with a low metal rim. Above the rim was a 30 cm tall transparent Plexiglas wall which enabled easy view outside of the arena. The arena was 1 m above the ground in a room with a sufficient amount of visual cues (doors, colored signs on walls, etc.), which served as orientation cues during behavioral testing. On the margin of the arena was a light-emitting diode (LED), which tracked rotation of the arena during the experiment. Another LED was used to track movement of a subject. This LED was fixed to a small metal plate which was attached on the subjects back with two rubber harnesses before each session. A cable for administering electric shocks ran to the metal plate. The cable was attached to a hypodermic needle implanted in a skin fold of the subject with an alligator clip. The position of a rat was tracked during the session with a sampling frequency of 25 Hz by a computer which was located in the neighboring room. A program used for tracking a rat’s position (Tracker 2.33; Biosignal Group, Brooklyn, New York, USA) simultaneously on-line evaluated whether the rat was within the to-be-avoided sector and administered a mild electric current in that case. Data were stored for off-line analysis which was conducted with Carousel Maze Manager 0.4.0 (Bahník, 2014). The electric current (AC, 50 Hz, 0.5 s) was administered whenever the subject entered the to-be-avoided sector for a duration longer than 300 ms. The administered current was initially adjusted to the rat’s reaction to elicit response but not a freezing response. All but two subjects responded to 0.4 mA, which was subsequently used for the rest of the experiment. The two mentioned subjects that did not respond to a current of any intensity (maximum used was 0.7 mA) were excluded from analysis (the exclusion is further described in Results). Whenever a subject did not escape the to-be-avoided sector within 900 ms of the previous shock, another was administered.

Figure 1 A photograph of the Carousel apparatus.

The apparatus consists of a metallic disk which can be rotated at various speeds and directions. A rat is connected with a wire to the swivel on the ceiling. The wire supplies the LED on the rat and delivers mild footshocks (0.2–0.7 mA).

Measured parameters

The following parameters were used for subsequent analyses: Total distance was computed as a sum of distances between positions of a subject within an arena (that is without displacement by rotation of the arena) sampled with frequency 1 Hz and was used to assess locomotion of subjects. Maximum time of avoidance was computed as the maximum duration between two subsequent occurrences of a subject in the to-be-avoided sector. This measure was used to estimate the ability to avoid the to-be-avoided sector. Maximum time of avoidance was equal to 1,200 s when a subject did not get any shock. Maximum time of avoidance usually highly negatively correlates with the number of received shocks. Its distribution is closer to normal and therefore is better suited for analysis of avoidance. Directional mean denotes the average direction of vectors from the center of the arena to subject’s position. This can be otherwise described as the direction of a vector obtained from summing unit vectors with directions equal to the direction of a subject relative to the center of the arena. The directional mean may be used to assess strategy of avoidance of the to-be-avoided sector. Circular variance denotes a variability of directions of vectors from the center of the arena to subject’s position. It is computed as one minus the length of the vector obtained by summing unit vectors with directions equal to the direction of a subject relative to the center of the arena divided by the number of these vectors. This measure shows to what degree the subject prefers a specific position within a room.2 Periodicity of movement is computed as the median duration of continuous intervals during which a subject is not moving. It may suggest a strategy used to avoid the to-be-avoided sector. Time in the adjacent sector shows the proportion of time which a subject spent in the sector adjacent to the to-be-avoided sector. The adjacent sector was chosen to be a section of the circle 60° wide, i.e., of the same width as the to-be-avoided sector. The center of the adjacent sector lay 60° against the rotation of the arena from the center of the to-be-avoided sector, i.e., the sector from which a subject is moved to the to-be-avoided sector in case of immobility. Median speed after shock was computed as the median angular velocity 1 s after shock which was not followed by another shock sooner than 1 s. Positive median speed after shock shows movement against the rotation of the arena. It may reveal whether a subject moves preferentially against or with the direction of rotation of the arena and with what speed.

Results

All analyses were done in R 3.0.2 (R Development Core Team, 2011). Analysis scripts as well as data and additional details of procedure are available on osf.io/z5dny/.

Learning phase

Two subjects that were not able to learn the task were excluded from analysis. These rats were removed from the experiment after the seventh day of the learning phase. Their visibly lower locomotion and maximum time of avoidance can be seen in Fig. 2 (see also Fig. 4B). Maximum time of avoidance was at a level corresponding to the absence of locomotion—near 50 s, which is a duration between two subsequent presences of a rat in the to-be-avoided sector in case of its immobility (when the arena rotation speed is 1 rpm).

Figure 2 Performance parameters.

Crosses denote mean values for a given group and session (two excluded subjects depicted by empty circles are not included in the mean). Abbreviations: LP, Learning phase; PP, Probe phase; DP, Dark phase (A) Total distance in meters. It can be seen that both groups did not show any difference in locomotion during learning and probe phases. Lower locomotion can be seen in one subject from the control group only during the first two days. However, it attained the level of other subjects by the third session. Absence of a difference between groups in total distance suggests that the experimental manipulation did not cause a higher requirement on locomotion in the experimental group. (B) Maximum time of avoidance in seconds. Maximal possible maximum time of avoidance was equal to the duration of a session, i.e., 1,200 s, which corresponds to an absence of entrances in the to-be-avoided sector. Five subjects from each group attained this time during the probe phase, which shows that subjects from both group learned the task well during the learning phase. It can be seen that there was no reliable difference between groups during the learning and probe phases. The horizontal dashed line shows 50 s, which corresponds to the maximum time of avoidance of a non-moving subject (for a speed of arena rotation 1 rpm). It is clear that both subjects that were excluded from analysis were not able to actively avoid the to-be-avoided sector. A large decrease in performance is visible for all subjects during the dark phase. One subject from the experimental group was able to avoid the to-be-avoided sector for 646 s during the third session of the dark phase.

Analyses were performed with multilevel linear regression with day and group as predictors. Polynomial contrasts were used for the day factor (Baguley, 2012). Only results for linear and quadratic contrasts are reported because higher order contrasts would be hard to interpret. The linear contrast accounts for linearly decreasing or increasing trend and the quadratic contrast accounts for U-shaped trend in data. In combination they fit well a large pattern of possible results.

Analysis for total distance did not suggest any group effect, t(12) = 0.58, p = .57, r = .16. Linear contrast for day was not significant, t(96) = 1.37, p = .17, r = .14, but quadratic contrast was, t(96) = − 2.69, p = .008, r = .26. The interaction of day and group factors was not significant either for linear, t(96) = − 1.31, p = .19, r = .13, or quadratic contrast, t(96) = 1.70, p = .09, r = .17. Therefore, data did not show substantial effect of experimental manipulation on locomotion. This suggests that the task had similar locomotor requirements for both groups. Results for total distance are displayed in Fig. 2A.

Maximum time of avoidance during the learning phase did not differ between groups, t(12) = 1.00, p = .34, r = .28. Change of performance between days was seen in both linear, t(96) = 6.10, p < .001, r = .53, and quadratic contrasts, t(96) = − 3.46, p < .001, r = .33.3 Neither the interaction between linear contrast for day and group, t(96) = − 0.80, p = .42, r = .08, nor between quadratic contrast and group, t(96) = 0.77, p = .44, r = .08, were significant. This shows that performance of both groups improved during the learning phase (see Figs. 4A and 4C for sample behavioral graphs). The difference between groups was seen neither in performance nor in speed of learning. Results for maximum time avoided are depicted in Fig. 2B.

Probe phase

The Wilcoxon test suggested that subjects in the experimental group stayed somewhat further away from the to-be-avoided sector than subjects in the control group, W = 11, p = .10, n1 = n2 = 7, r = .46. Values of directional means for individual subjects are displayed in Fig. 3A.

Figure 3 Strategies parameters.

Crosses denote mean values for a given group and session (two excluded subjects depicted by empty circles are not included in the mean). Abbreviations: LP, Learning phase; PP, Probe phase; DP, Dark phase (A) Directional mean. Possible values of directional mean range from 0° to 360°. The value of 0° and 360° correspond to the same direction. The to-be-avoided sector is indicated by the dashed line and spans the values of 0° to 60°. The arena rotation was towards the lower values, which means that larger values of directional mean show that a subject was further away from the to-be-avoided sector. A somewhat higher average directional mean for the experimental group than for the control group can be seen during the probe phase. Five subjects with the highest values of directional mean in this phase were from the experimental group. The higher directional mean suggests that subjects in the experimental group used a safe strategy which was based on a preference of places more distant from the to-be-avoided sector in terms of time required for movement of the subject to the to-be-avoided sector by arena rotation. (B) Time in the adjacent sector as a proportion to the length of a session. The dashed line shows a proportion of time corresponding to homogenous distribution of presence of a subject in all sectors of the arena, which would be expected during full inactivity of the subject. It can be seen that the two subjects excluded from analysis are close to this line. Time in the adjacent sector decreases during the learning phase and is close to zero for nearly all rats at its end. This shows that subjects do not learn only learn to avoid the to-be-avoided sector, but their presence is getting more sparse in the adjacent sector as well. This behavior may be advantageous if subjects do not form an exact representation of the position of the to-be-avoided sector within the room. Avoidance of a wider sector may thus lead to better performance in the task. Large increases in time in the adjacent sector can be seen during the dark phase. Time in the adjacent sector in the dark phase often exceeds values expected in cases of homogenous distribution of presence in all sectors of arena. The reason for this is that subjects avoided the to-be-avoided sector with movement against the direction of rotation, but this movement was often initiated only after the administration of a shock. (C) Circular variance. Circular variance has a range of values from 0 (a subject is present only in one direction with relation to the center) to 1 (homogenous presence of a subject in all directions in relation to the center—circular variance reaches this value in the case of an immobile subject). In the text, circular variance is analyzed only for the probe phase where similar values can be seen for both groups. Decreasing circular variance can be seen during the first days of the learning phase. This development show that subjects learn to move within a restricted area of the room during the first few days. While the figure suggests a difference between the experimental and control groups in the learning phase, this difference is hard to interpret because it can be caused by the different speed of arena rotation, and not by any difference in the behavior of subjects. High circular variance can be seen in subjects excluded from analysis. It stems from an absence of active avoidance of the to-be-avoided sector. Low circular variance of one subject in the experimental group on the second day of the dark phase can be ascribed to its reaction to the movement into the to-be-avoided sector, which consisted of a short movement against rotation of the arena. Since the rat traveled only a short distance this way, it moved within a narrow sector of the arena which was near the to-be-avoided sector. This is also the cause of its 68 entrances into the to-be-avoided sector and 78.3% of time in the adjacent sector (see Fig. 3B).

Even though a strong correlation of time in the adjacent sector and directional mean suggests that both parameters measure a similar construct, rS(12) = − .57, p = .03, no difference between groups was found for time in the adjacent sector, W = 24, p = 1, n1 = n2 = 7, r = .02. A possible reason may be that values for time in the adjacent sector were low for both groups. Four subjects from the experimental group and three from the control group spent less than 0.5‰ of time in the adjacent sector. The absence of difference might have been easily a result of the floor effect. Values of time in the adjacent sector are depicted in Fig. 3B.

The Wilcoxon test showed no difference in circular variance between groups, W = 30.5, p = .48, n1 = n2 = 7, r = − .21. Circular variance for both groups can be seen in Fig. 3C.

The Wilcoxon test for periodicity of movement did not reveal any effect of experimental manipulation, W = 24.5, p = 1, n1 = n2 = 7, r = 0. Periodicity of movement in the probe phase correlated significantly with circular variance, rS(12) = .73, p = .003, which confirms that periodicity of movement and circular variance measure a similar characteristic of subject’s behavior in the task. Higher periodicity of movement means a longer distance that the subject moves by arena rotation during its inactivity. Since subjects usually move only in a certain sector within a room, subjects correct their movement caused by the arena rotation by returning to the position where their inactivity began. Higher periodicity of movement thus causes subject’s presence in a wider arena sector within the room frame and therefore leads to higher circular variance as well. The association between periodicity of movement and circular variance indicates validity of both these measures and supports their usability for testing of specific hypotheses in future research.

Dark phase

Results for maximum time of avoidance can be seen in Fig. 2B. The Wilcoxon test did not reveal a significant difference in the average maximum time of avoidance between experimental (M = 153 s, SD = 90 s) and control (M = 110 s, SD = 25 s) groups, W = 18, p = .44, n1 = n2 = 7, r = .22. Higher mean and larger variability of values of the experimental group are caused primarily by performance of one subject (hereafter referred as rat 15) with the average maximum time of avoidance 354 s.4

Although performance during the dark phase worsened considerably in comparison to performance in learning and probe phases, some ability to avoid the to-be-avoided sector was visible even in the dark phase (Fig. 4D). A half of the subjects had at least one value of maximum time of avoidance higher than 175 s. Since an absence of locomotion leads to maximum time of avoidance 50 s, these values show that the subjects had to actively avoid the to-be-avoided sector for, at least, two minutes. The highest measured value of maximum time of avoidance was for rat 15, that was able to avoid the to-be-avoided sector during the third day of the dark phase for 646 s, more than half of the duration of the session (Fig. 4E).

Figure 4 Sample angular position graphs.

The graphs present positions of subjects in relation to the to-be-avoided sector (shown by red horizontal lines) during a given sessions. Animals tend to move at the periphery of the arena which means that the angular position displayed in the figure is usually sufficient to represent the exact position of a subject within a room. Only rarely do they move in the center of the arena (indicated by wheat color of vertical bars). Subjects are immobile for most of the session (white) and their displayed movement during this time is only due to the rotation of the arena. The to-be-avoided sector is usually avoided by movement against the direction of the rotation (light green) and when subjects receive a shock (shown as red ticks below a graph) they tend to leave the sector in the same direction (dark green). Movement in the direction of arena rotation (light blue) is present in the initial sessions (A), but usually disappears in the subsequent sessions. The movement in the direction of arena rotation is sometimes used to escape the to-be-avoided sector after a shock (dark blue) especially in subjects who do not learn to actively avoid the sector (B). In the initial sessions, subjects sometimes do not react to a shock by active movement (crimson). (A) First session of the learning phase. The subject did not have any experience with the task which can be seen from the wide range of displayed behaviors. A sucessful strategy of avoidance predominates at the end of the session. (B) Seventh session of an excluded subject. It can be seen that the subject does not actively avoid the to-be-avoided sector. When taken into the sector by arena rotation, it usually escapes further shocks by movement in the direction of the rotation. (C) Ninth session of the learning phase. The subject learned to successfully avoid the to-be-avoided sector for the whole session by movement against the direction of arena rotation. (D) Third session of the dark phase of a rat with bad performance. While the subject still moves against the direction of arena rotation, it cannot navigate using external cues, which means that it cannot regulate its position within the room properly to avoid entrance of the to-be-avoided sector. Movement is often initiated only after receiving a shock. (E) Third session of the dark phase of a rat with good performance. Rat 15 was able to avoid the to-be-avoided sector for 646 s during the displayed session (4:11–14:57). It can be seen that it was able to avoid the sector better than during the first session of the learning phase (A), but worse than during a later session of the learning phase (C). Its position within the room was not as stable as during the ninth session of the learning phase and it was sometimes regulated only after receiving a shock.

Even though rats were somewhat able to avoid the to-be-avoided sector, their performance did not improve during the three sessions. Subtracting maximum time of avoidance for the first day from the value for the third day results in positive values only for 5 out of 14 subjects. The average of these values was 3 s, which shows that with the exception of rat 15, which improved between the first and the third day by 444 s, rats were not improving during the dark phase.

Although subjects were not generally able to avoid the to-be-avoided sector for an extended period of time, basic avoidance behavior was observed even in this phase. This can be seen from the positive values of median speed after shock (analyzed for both groups together), which were higher than zero for all three days of dark phase, ts(13) > 4.11, ps < .002, 24° /s <Ms < 28°/s. This shows that subjects avoided the to-be-avoided sector predominantly by movement against the direction of rotation of the arena, i.e., in a similar manner they solve the task in light (Fig. 4D).

Discussion

The results did not show an effect of variable arena rotation speed on locomotion or the ability to avoid the to-be-avoided sector in the learning phase. Subjects from both groups were able to successfully learn the task and their performance was relatively stable from the fifth day of the learning phase. Therefore, the experiment does not suggest that stable arena rotation speed helps subjects to learn the task.

If subjects used a temporal strategy for avoidance, we would expect that use of this strategy is easier when arena rotation speed is stable rather than variable. We would therefore expect better performance of the control group. Since no difference in performance between the experimental and control group was found, we can conclude that a temporal strategy is not necessary for solving the task. While we can say that a temporal strategy is not necessary for solving the task, it cannot be conclusively inferred from the results that subjects do not use a temporal strategy. It is possible that subjects in the experimental group compensated for the inability to use a temporal strategy by using a different strategy of avoidance. The result is nevertheless important because it shows that a deficit of interval timing should not by itself lead to worse performance in the task. For example, if it is known that some drug causes deficits in interval timing, the results of this experiment suggests, that its possible effect on performance in the APA task could not be assigned only to this effect of the drug.

Subjects in the experimental group may use two strategies to be safe from being moved into the to-be-avoided sector by faster rotation of the arena if we assume that they move against the rotation of the arena with a certain periodicity. The strategies were assessed in the probe phase. The first possible strategy is to stay further away from the to-be-avoided sector; this ensures that even faster rotation of the arena does not move them into the to-be-avoided sector during the period of inactivity. Consistently, a higher directional mean was seen for the experimental group, which suggests that the subjects moved in positions further from the to-be-avoided sector. It should be noted that while this result was hypothesized, it did not reach statistical significance and is based on a small sample of subjects, so caution with regards to conclusions from it is warranted. The second possible strategy is to move with a lower periodicity. This strategy would enable subjects in the experimental group to adjust their position in the room more often and would again prevent them from being moved into the to-be-avoided sector even during faster arena rotation speed. However, we found no difference which would suggest employment of this strategy in circular variance and periodicity of movement between the groups. It is possible that the rats from the experimental group used a strategy indicated by higher directional mean, i.e., they avoided being moved into the to-be-avoided sector by being further away from it and not by moving within a narrower sector, which would be suggested by a difference in circular variance or periodicity. Both strategies are not mutually incompatible, but using one of them may be sufficient for successful avoidance of the to-be-avoided sector even when the speed of arena rotation is higher.

In addition to suggesting a strategy used for avoidance of the to-be-avoided sector, the results of the probe phase showed the possibility of using new parameters to assess specific hypotheses about the influence of experimental manipulations on behavior in the task. From a positive correlation between circular variance and periodicity of movement, it can be seen that both parameters measure a similar construct which partially validates both parameters. Similarly, a negative correlation between directional mean and time in the adjacent sector suggests a convergent validity of both parameters.

Since allothetic avoidance cannot be used in dark, any possible difference between groups in the dark phase may reveal the effect of manipulation on the use of alternative strategies of avoidance. However, the results of the dark phase did not suggest any difference between the experimental and control group. Both groups considerably worsened in comparison to the probe phase. Nevertheless, maximum time of avoidance and median speed after shock showed that subjects were still able to solve the task to a certain degree. Movement against the direction of arena rotation, which keeps a subject outside of the to-be-avoided sector by correcting its position within a room (Stuchlik et al., 2013), persisted even in the dark in most of the subjects. The length of this movement could not have been adjusted by cues outside of the arena and therefore subjects were often moved by arena rotation into the to-be-avoided sector. With the exception of one subject, we did not observe any evidence of improvement of performance over time. It cannot be ruled out that learning avoidance in dark requires a qualitative change of strategy of avoidance and that we would observe an improvement even in other subjects if they had more time for learning. The general lack of improvement cannot be explained by limits of accuracy of idiothetic orientation because one rat was able to avoid the to-be-avoided sector for more than ten minutes. Idiothetic orientation is therefore sufficient for solving the task in the dark, but it does not seem to be crucial for solving the task in light. Since the subject that was able to avoid the to-be-avoided sector in dark was from the experimental group, it seems that solving the task in the dark may be based on a combination of inertial and substratal idiothesis rather than on interval timing and substratal idiothesis. A cue indicating when to be active would thus stem from information about passive movement in space rather than from time that passed since the previous movement. Of course, it cannot be excluded that both of these sources of information are combined during the task in dark. It is also possible that the subject was able to learn a temporal strategy during the probe and the dark phases where the speed of arena rotation was stable. Possible conclusions from the results of the dark phase are necessarily limited by the fact that only one of the subjects was able to reliably avoid the to-be-avoided sector (Fajnerova et al., 2014).

The most important limitation, which restricts reliability of conclusions from the study, is a relatively low number of subjects. It is possible that any absence of difference between groups was caused by small statistical power stemming from a low number of subjects in both groups (Button et al., 2013). However, some results are not limited by this issue. For example, it can be seen that even subjects in the experimental group were able to quickly learn the task in light. Even though it is possible that some difference would be found if the experiment had higher statistical power, it is clear that stable arena rotation speed is not necessary for learning the task. Furthermore, it can be seen that rats can learn the task in the dark, even if only one was able to do so. This finding is consistent with the paper by Fajnerova et al. (2014). Usefulness of some of the newly presented parameters is also clear from their mutual associations that we observed in the present study.

An additional possible limitation is the relatively small range of speeds of arena rotation in the experimental group. The highest speed was only one third faster than the speed used in the control group. This limited range of speeds was due to the technical characteristics of the experimental apparatus. Although it is possible that a wider range of arena rotation speeds would lead to a difference between groups, it is not clear whether a higher speed of arena rotation would not lead to different effects than those that were studied in the present experiment. The goal of the experiment was not to explore the influence of arena rotation speed itself, but of its variability. The arena rotation speeds that we used varied between each session and the difference between maximum and minimum speed was 0.74 rpm. This was considered to be sufficient to make time an invalid cue for when a subject should move.

We would like to thank Michaela Fialová, Jindřich Kalvoda and Antonín Zahálka for their technical assistance and Peter M. Luketic for proofreading.

Abbreviations

APA active place avoidance

rpm revolution per minute

Additional Information and Declarations

Competing Interests

Author Contributions

Animal Ethics

Data Availability

1 At the end of the dark phase, one day was added for 6 selected rats that showed some ability to avoid the to-be-avoided sector in the dark. During this day, one session of the APA task in the dark took place to clarify obtained results. The results are described in Supplementary materials on osf.io/z5dny/.

2 For better understanding, directional mean and circular variance can be illustrated using sample tracks in Fig. 4. It is possible to see that a subject depicted in Fig. 4C moved within a narrow sector of the arena. Its circular variance is correspondingly .20. On the other hand, the subject depicted in Fig. 4B showed little preference for a specific sector and its circular variance is thus .86 (i.e., close to 1). The subject depicted in Fig. 4C also moved further away from the center of the arena (in terms of angular movement required for its passive displacement to the to-be-avoided sector by rotation of the arena). Correspondingly, its directional mean is 192.9°, while subjects moving closer to the to-be-avoided sector have lower directional means such as 60.1° (Fig. 4D) and 72.1° (Fig. 4E).

3 A model defined by significant independent variables (intersection with ordinate, linear and quadratic contrasts for day) shows improvement during time which was initially faster, was subsequently getting slower, and reached a peak during the seventh day (prediction of a model is 1,021 s). However, from the fifth to the last day, the predicted maximum time of avoidance is within a narrow range 951–1,021 s. Attainment of this relatively stable level can be seen in Fig. 2B.

4 The mean and standard deviation for maximum time of avoidance of the experimental group are 120 s and 19 s without this subject.

The authors declare there are no competing interests.

Štěpán Bahník conceived and designed the experiments, performed the experiments, analyzed the data, contributed reagents/materials/analysis tools, wrote the paper, prepared figures and/or tables.

Aleš Stuchlík conceived and designed the experiments, wrote the paper.

The following information was supplied relating to ethical approvals (i.e., approving body and any reference numbers):

Committee on the Ethics of Animal Experiments of the Institute of Physiology Academy of Sciences of the Czech Republic (Permit Number: 136/2013).

The following information was supplied regarding data availability:

Open Science Framework (osf.io/swxhb/).

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
