# Peer review of "Temporal and spatial strategies in an active place avoidance task on Carousel: a study of effects of stability of arena rotation speed in rats"

_PeerJ, doi:10.7717/peerj.1257_

## Round 0.1 · original submission · Major Revisions

After careful consideration, I feel that your manuscript has merit, but is not suitable for publication as it currently stands. Therefore, my decision is "Major Revision." I invite you to submit a revised version of the manuscript that addresses all points raised by the reviewers. In particular, reviewer 1 felt that more positive results should be reported. Whilst negative or inconclusive results are acceptable following PeerJ criteria, I agree with the reviewer that this study would be significantly enhanced if the number for animals evoking positive results is increased. In addition, the paper is far too long for a study with limited positive data. Please shorten if possible.

Reviewer 1 ·

Basic reporting

This is an interesting article dealing with the performance of rats placed in a round and rotable dry area. The task is designed for determining temporal and spatial strategies in behaving animals under different experimental situations.

Experimental design

In general, experiments are well designed and performed and the paper is well written and documented. Nevertheless authors should address the comments and suggestions listed below.

Validity of the findings

The main problem of this paper is that most of the experiments included generated negative results. In this situation, the text is too much long and redundant. In this regard, the text should be trimmed as much as possible.

Apparently, the main positive result presented in this study is referred to data collected from just one rat. If possible, authors should try to increase the number of animals evoking this positive results related to animals performance in the Carrousel during the dark phase of the task.

The main conclusion/s of the study should be presented in a more clear way.

The writing should be improved. For example the word “task” is repeated three times in the two first lines of the Abstract.

Additional comments

See above

·

Basic reporting

This is an interesting paper investigating the effects of revolution speed on performance in APA. The graphical representation of the data is outstanding. The authors should address the following suggestions and concerns.

Clear definitions of allothetic and idiothetic orientation are required.

It looks like the report is showing no difference due to the current treatments. How was the variable rotation speed group varied? Some description is provided but was there a set combination order of changes? Do any reach 0.6 or 1.34 RPMs?
What if the limits were pushed beyond 0.7 or 1.3 RPMs? Is the current design better than setting speed at different levels (e.g., 0.5, 1.0, 1.5. 2.0 RPMs)? This would show increased differences and tell us much more about some of the measures. It would also be interesting to see these type group comparisons in the PP and DP phases (after a delay of testing but keeping the original RPM differences the same during testing).

Directional means and circular variance could be more easily understood with a schematic. These could be a part of Figure 1 and represented on a representation of the rotating metallic disk (or better yet “arena”). Where are the actual indices of error measurements (i.e., shocks) in this work? Shocks would be interested to most readers and it is not provided. Can shocks be included?

In Figure 3B, Time in the adjacent sector is in sec, right? Can you include this in the label? What are the measurement units for Circular Covariance in part C?

Figure 4 is a little confusing. It is not intuitive and needs much additional explanation. A full description of the excursions becoming less with training and thus not exceeding the error points for shock (i.e., the high and low horizontal line) and some additional time on the color coding that represents events and/or behaviors would help focus the reader’s attention and understanding. When do the shocks/errors occur (red)? Maybe it would be more clear if the figure focuses on the first and ninth session for the two groups in more detail first (i.e., most important in the goals of the study) then go to the Dark (non-solver vs. solver). The dark solver here actually looks pretty good in spite of the complete drop in performance in earlier graphs. This might need a comment about it being an outlier and certainly not equivalent to a 9th session performance of a representative rat in the light (above in the figure). As a side, the terms dark solver and non-solver are not clear initially. This could be communicated more clearly.

Experimental design

approriate; see above comments

Validity of the findings

approriate; see above comments

Additional comments

approriate; see above comments

---

## Round 0.2 · accepted · Accept

The paper has been greatly improved and no further revisions are needed.

Reviewer 1 ·

Basic reporting

I think that authors have addressed all of my comments and suggestions. I hace no further comments.

Experimental design

See above.

Validity of the findings

See above.

Additional comments

See above.

·

Basic reporting

No further comments

Experimental design

no further comments

Validity of the findings

no further comments

Additional comments

no further comments